# Neurocomputational Underpinnings of Suboptimal Beliefs in Reinforcement Learning Agents

**M Ganesh Kumar**[1*], **Adam Manoogian**[2*], **Billy Qian**[1], **Cengiz Pehlevan**[1†], **Shawn A. Rhoads**[3†]

[1]Harvard University, [2]Monash University, [3]Icahn School of Medicine at Mount Sinai

## Abstract

**Maladaptive belief updating is a hallmark of psychiatric disorders, yet its underlying neurocomputational mechanisms remain poorly understood. While Bayesian models characterize belief updating in decision-making, they do not explicitly model neural computations or neuromodulatory influences. To address this, we developed a recurrent neural network-based reinforcement learning framework to investigate decision-making deficits in psychiatric conditions, using schizophrenia as a test case. Agents were trained on a predictive inference task commonly used to assess cognitive deficits found in schizophrenia, including under-updating beliefs in volatile environments and over-updating beliefs in response to uninformative cues. The task thus included two conditions: (1) a change-point condition requiring adaptation in a volatile environment and (2) an oddball condition requiring resistance to outliers. We modeled these deficits by systematically manipulating key hyperparameters associated with specific neural theories: reward prediction error (RPE) discounting and scaling (reflecting diminished dopamine responses), network dynamics disruption (reflecting impaired working memory), and rollout buffer size reduction (reflecting decreased episodic memory capacity). These manipulations reproduced schizophrenia-like decision-making impairments and revealed that suboptimal agents exhibited fewer unstable fixed points near network activity in the change-point condition, suggesting reduced computational flexibility. This framework extends computational psychiatry by linking cognitive biases to neural dysfunction and provides a mechanistic approach to studying decision-making impairments in psychiatric disorder.**

**Keywords:** computational psychiatry; recurrent neural network; reinforcement learning; decision-making; schizophrenia; belief inference

## Introduction

The field of computational psychiatry has made significant progress in characterizing neural and cognitive deficits in psychiatric disorders (Wiecki et al., 2015; Huys et al., 2016; Bennett et al., 2019; Rhoads et al., 2024). However, current theoretical models often separately describe behavior or neural activity, which limits their ability to uncover mechanistic links between the two. Schizophrenia (SCZ), for example, is associated with many impairments including positive symptoms of delusion as well as cognitive impairments such as inaccurate belief inference—patients struggle to integrate past experiences with new evidence to guide decision-making (Dudley

et al., 2016; Baker et al., 2019; Bronstein et al., 2019; Sheffield et al., 2023; Karvelis et al., 2023). Although many Bayesian and reward prediction learning models have traditionally captured behavioral accounts of these deficits (Gibbs-Dean et al., 2023; Katthagen et al., 2022), they do not capture how neural dynamics give rise to impaired inference. Recurrent neural networks (RNNs) trained with reinforcement learning (RL) algorithms offer a promising alternative (Wang et al., 2018), as meta-RL agents can learn task representations from experience while simultaneously generating emergent neural activity patterns that can be analyzed to reveal the underlying computational mechanisms.

Here, we train RNN-RL agents to perform a predictive inference task commonly used to assess cognitive deficits in SCZ (Nassar et al., 2021). By manipulating key hyperparameters linked to specific theories of neural underpinnings, we generate near-optimal (e.g., healthy controls) and suboptimal (e.g., SCZ) behavior, allowing us to compare the agents' performance with empirical patient data from Nassar et al. (2021). Importantly, we leveraged dynamical systems analysis to examine the internal neural dynamics of the RNN (Sussillo & Barak, 2013) to identify how disruptions in stability contribute to maladaptive belief inference. Our results demonstrate that suboptimal agents lack fixed points, mirroring neural dysfunction observed in SCZ. These findings highlight the utility of RNN-RL models in linking behavior with neural mechanisms, providing a more integrated approach to computational psychiatry.

## Related Works

### Cognitive models of maladaptive belief inference

Bayesian models provide a useful framework for understanding uncertainty reduction in decision-making (Behrens et al., 2007), but their predictive power is often limited to specific tasks (Loosen et al., 2024). SCZ is broadly associated with impairments in context-dependent decision-making, though these deficits are highly task-sensitive (Ravizza et al., 2010; Kaplan et al., 2016; Okruashvili et al., 2023; Choung et al., 2022). Cognitive biases, such as reduced information integration, increased reliance on prior beliefs, and resistance to disconfirmatory evidence, have been proposed as key contributors to SCZ-related decision-making impairments (Dudley et al., 2016; Baker et al., 2019; Kirschner et al., 2024; Sanford et al., 2014; Sterzer et al., 2019) and are thought to underlie core symptoms of SCZ, including delusional beliefs and cognitive rigidity (Bronstein et al., 2019; Sheffield et al., 2023).

For instance, in a predictive inference task developed by Nassar et al. (2021), individuals with schizophrenia exhibited

two distinct behavioral phenotypes. In change-point conditions which require heightened adaptation to new volatile evidence, these individuals tend to **under-update** their beliefs. Conversely, in oddball conditions, they display a propensity to **over-update** to uninformative, noisy stimuli, resulting in less precise beliefs. Two separate normative models, based on Bayesian updates, were applied to elucidate these phenotypes in patient behavior. However, these models have several limitations, including poor reliability (Loosen et al., 2024) and a lack of predictions about underlying neurocomputational underpinnings. Thus, a neural network-based model could provide valuable insights into computational processes and address these limitations.

## Neural models of maladaptive belief inference

Mechanistic theories suggest that belief inference deficits could arise from disruptions in several neurobiological processes. The **dopamine hypothesis** posits that hyper-activity in the mesolimbic pathway contributes to positive symptoms that lead to delusions or hallucinations, while hypo-activity (blunted dopamine responses) disrupts cognitive control, including reward processing and working memory (Juckel et al., 2006; Maia & Frank, 2017; Meyer-Lindenberg et al., 2005; Gradin et al., 2011). Some studies suggest that the reward sensitivity itself may be intact, but SCZ patients fail to integrate information when calculating the value of possible decisions (Heerey et al., 2008). Beyond reinforcement learning deficits, **disruptions in working memory function** also negatively impact decision-making flexibility (Loh et al., 2007; Rolls et al., 2021). **Episodic memory dysfunction** may also alter the retrieval of past experiences, leading to a hyperreliance on immediate stimuli and reduced learning from previous outcomes (Achim & Lepage, 2005; Danion et al., 2007; Ashinoff & Horga, 2020).

## Methods

### Predictive Inference Task

We implemented the predictive inference decision-making task from Nassar et al. (2021), in which an agent predicts and catches a falling bag to receive a reward. The bag is dropped from a helicopter positioned randomly between 0 and 300 units, following a Gaussian distribution centered at the helicopter's location (SD = 20 units). On each trial, the agent moves a bucket in 10-unit increments and confirms its position to trigger the bag drop. Rewards are based on the accuracy of the catch, with feedback provided as the prediction error (i.e., the difference between the bag's actual and predicted landing locations).

During the training phase, the helicopter's location is provided to the agent, and the optimal strategy is to place the bucket directly beneath the helicopter to maximize the average reward. In the evaluation phase, the helicopter's location is hidden, and the agent must infer the bucket placement based on the evidence from previous bag drops.

The task includes two conditions that affect the underlying statistics of the task environment:

1. **Change-Point (CP) Condition**: The helicopter's location changes to a new position sampled uniformly between 0 and 300 units, with a hazard rate of 0.125. The optimal solution is to rapidly adjust the bucket position when the bag drop location changes, as this indicates a shift in the helicopter's location.

2. **Oddball (OB) Condition**: The bag drop location is sampled from a random uniform distribution between 0 and 300 units, deviating from the helicopter's location, with a hazard rate of 0.125. In this condition, the helicopter's location follows a random walk following a Gaussian distribution with a standard deviation of 7.5 units. The optimal solution is to ignore outlier bag drops that deviate significantly from the expected location and maintain the bucket position.

### Network Architecture

Figure 1a illustrates the network architecture of our agents. The input to the network consists of the same information provided to human subjects in Nassar et al. (2021), which includes: 1) the bucket location at each time step in the current trial, 2) the bag drop location in the previous trial, 3) the prediction error, which is the difference in location between the previous trial's bag drop location and the chosen bucket location, and 4) the helicopter's position, which is only provided during the training phase.

Additionally, the task condition—either the change-point or oddball condition—is encoded as a one-hot vector, similar to the instructions given to human subjects in the original study. The observation vector $\boldsymbol{o}_t$, context vector $\boldsymbol{c}$ and the reward obtained $r_t$ are passed as inputs to a recurrent neural network (RNN) with Tanh nonlinearity. The RNN is defined as:

$$h_i(\boldsymbol{o}_t) = \text{Tanh}\left(\sum_k^D W_{ik}^o \boldsymbol{o}_t + \sum_j^N W_{ij}^h \boldsymbol{h}_{t-1} + w_i^r r_t\right), \quad (1)$$

$$v(\boldsymbol{o}_t) = \sum_i^N w_i h_i(o_t), \quad (2)$$

$$a_j(\boldsymbol{o}_t) = \text{softmax}\left(\sum_i^N W_{ji} h_i(o_t)\right), \quad (3)$$

where $h_t$ is the hidden state of the RNN, which is passed to a scalar critic $v_t$ to compute the value function, and to three actor $a_t$ units ($M = 3$), which map to three possible actions: move left, move right, or confirm bucket placement. The agent uses a stochastic policy and samples a discrete actions from the probabilistic distribution $a_t$ (Wang et al., 2018; Lin et al., 2023). The agent can repeatedly move to the left or right (in increments of 10 units) before confirming the bucket placement, which initiates the bag drop, disburses the reward, and starts the next trial. The use of a recurrent neural network is motivated by its ability to maintain and integrate temporal information, making it suitable for solving Partially Observable

Markov Decision Process (POMDP) tasks (Singh et al., 2023). Furthermore, this architecture enables trained agents to adapt to novel task conditions without gradient optimization, demonstrating a form of meta-learning (Wang et al., 2018).

## Model Optimization

Agent weights were optimized using the Advantage Actor-Critic (A2C) algorithm (Mnih et al., 2016; Kumar et al., 2021; Lin et al., 2023; Kumar & Pehlevan, 2024). The objective function combines the policy gradient and the value function loss (Wang et al., 2018; Kumar et al., 2024), defined as:

$$\nabla \mathcal{L} = \nabla \mathcal{L}_\pi + \nabla \mathcal{L}v$$
$$= \nabla_\theta \log \pi(a_t | \boldsymbol{o}_t; \theta) A_t(\boldsymbol{o}_t; \theta_v)$$
$$+ \nabla_{\theta_v} v(\boldsymbol{o}_t) A_t(\boldsymbol{o}_t; \theta_v), \quad (4)$$

$$A_t(\boldsymbol{o}_t; \theta_v) = \beta_\delta \left( \sum_{k=0}^{\tau} \gamma^k r_{t+1+k} - v_t(\boldsymbol{o}_t; \theta_v) \right), \quad (5)$$

where $A_t$ is the advantage function computed by taking the difference between the discounted return $\sum_k^\tau \gamma^k r_{t+1+k}$ and the value estimation by the critic $v_t$. Free hyperparameters for our modified reinforcement learning algorithm include include: 1) $\gamma$ the reward discount factor, 2) $\beta_\delta$ scales the Advantage function, and 3) $\tau$ is a lower bound on the minimum rollout length before computing the Advantage function. The model parameters are updated via gradient descent to maximize rewards, which has been shown to recapitulate several experimentally observed neural phenomena (Kumar et al., 2024).

Each epoch consists of 200 trials, alternating between change-point and oddball task conditions, with the starting state randomized. Agents were trained for 25,000 epochs where the helicopter's location was visible. Subsequently, they were trained for another 25,000 epochs in the evaluation phase, where the helicopter's location was hidden. Thereafter, the agents' weights were fixed, and their behavior were sampled over several epochs with different helicopter location initializations to reduce sampling error.

## Mechanisms to Influence Behavior

The RNN-RL agent has four free hyperparameters ($\gamma$, $\beta_\delta$, $p_{reset}$, $\tau$) that can be varied to modulate decision-making behavior. We varied these hyperparameters to simulate potential deficits linked to specific theories of neural underpinnings: reward prediction error (RPE) discounting, RPE scaling (mimicking blunted dopamine responses), working memory function (via network dynamics disruption), and episodic memory capacity (via rollout buffer size).

**Dopamine hypothesis** First, we consider the Advantage learning signal Eq. 5 to model dopamine responses, which has been shown to resemble the reward prediction error (Schultz et al., 1997; Montague et al., 1996; Starkweather & Uchida, 2021; Gershman & Uchida, 2019; Amo et al., 2022). The first hyperparameter is the reward discount factor (**1:** $\gamma$), which ranges from 0 to 1. A value closer to 1 places more weight on rewards obtained later in the trajectory, while a

value closer to 0 tends to ignore later rewards, especially if the trajectory is long. In other words, a smaller $\gamma$ places more emphasis on immediate rewards, potentially overlooking the cumulative rewards that can be maximized over time. The second hyperparameter scales the advantage signal (**2:** $\beta_\delta$) and ranges from 0.25 to 1.5, with 1.0 representing the unscaled learning signal.

**Working Memory disruption** The third hyperparameter is where we introduce Gaussian noise sampled from a distribution with mean 0 and standard deviation $1/\sqrt{N}$ into the recurrent activity ($h(o_t)$) based on a hazard rate (**3:** $p_{reset}$) to simulate noisy computation or random resets in neural dynamics Stein et al. (2021).

**Episodic memory capacity** The fourth hyperparameter is the size of the rollout buffer (**4:** $\tau$). The state, action, reward transitions $s_0, a_0, r_1, s_1, a_1, ...$ are stored in a memory buffer. When the buffer size (rollout length) reaches a minimum size of $\tau$, the reward-maximizing gradients (Eq. 4) are computed and the model weights are updated. A larger buffer computes the expected cumulative discounted reward over several trials, while a shorter buffer computes it over one or two trials. Additionally, agents with a large $\tau$ must sample more trials before updating their parameters and agents with a small $\tau$ must update their parameters more frequently, which can lead to exploding gradients. To account for this, we scale the learning rate based on the rollout size: $\eta = \eta_0\tau$, where $\eta_0 = 10^{-6}$. We vary the memory buffer size to assess the impact of recency biases on task performance Ashinoff & Horga (2020).

## $\Delta$Area Metric Quantification

We analyzed adaptive behavior using a novel $\Delta$Area metric comparing learning dynamics between conditions (Fig. 1D). For each trial $T$:

$$PE_T = H_T - B_T \quad \text{(Bag vs. bucket prediction error)} \quad (6)$$
$$UP_T = B_T - B_{T-1} \quad \text{(Bucket position update)} \quad (7)$$
$$LR_T = UP_T / PE_T \quad \text{(Learning Rate, normalized)} \quad (8)$$

The condition-specific **Area** was computed as $\sum_{i=1}^{200} LR_i \times PE_i$, analogous to ROC-AUC Fawcett (2006), quantifying how strongly agents adapt to prediction errors. The key metric $\Delta A = \text{Area}_{CP} - \text{Area}_{OB}$ evaluates optimal adaptation by contrasting change-point (high $\text{Area}_{CP}$ indicating appropriate updating) and oddball (low $\text{Area}_{OB}$ reflecting noise ignoring) conditions. Higher $\Delta A$ values indicate better discrimination between relevant and irrelevant information, providing a robust single metric for assessing hyperparameter effects across conditions.

## Fixed Point Analysis

We characterized the autonomous dynamics of trained RNNs by identifying fixed points in the vicinity of task-engaged hidden states. For each context, we clamped inputs to the relevant context cue (setting all other observables to 0) and minimized the squared dynamics speed $q(\mathbf{h}(o)) = \|\mathbf{F}(\mathbf{h}(o)) - $

$\mathbf{h}(o)\|^2$ using autodifferentiation methods Golub & Sussillo (2018), where $\mathbf{F}(\mathbf{h})$ represents the recurrent dynamics (Eq. 1). Starting from 2000 initial states sampled across 10 epochs (200 trials each, with helicopter/bucket/bag positions drawn from uniform/Gaussian distributions), we identified fixed points $\mathbf{h}^*$ satisfying $q(\mathbf{h}^*) = 0$. Stability was determined via Jacobian eigenvalue analysis ($|\lambda_i| < 1$ indicating stability). To assess the dynamical role of unstable fixed points, we computed the cosine similarity between their leading Jacobian eigenvectors ($v_1^*$) and actual state updates ($h_{t+1} - h_t$).

## Results

In this section, we study the bucket updating behavior of agents trained with different hyperparameters, analyze their recurrent neural network's dynamics, and compare our behavior metric against human subject data.

### Agents learn to predict bag drop locations

Figure 1B shows examples of behaviors exhibited by fully trained agents during the Change-point and Oddball conditions. These agents were initialized with different reward discount hyperparameters ($\gamma = 0.95$, top) and ($\gamma = 0.5$, bottom), representative of the high and low end ranges of the parameters tested. The agent initialized with a larger reward discount hyperparameter correctly adapts to prediction error, shifting the bucket to the most likely bag drop position in the next trial during the change-point condition. Importantly, the agent does not shift the bucket to outlier bags during the oddball condition. The agent initialized with a smaller reward discount hyperparameter also adapts the bucket prediction to the shifting bag drop during the change-point condition. However, this agent shows a tendency to over-update the bucket's location in the oddball condition, in an attempt to catch the outlier bags.

Figure 1C shows two agents' bucket update behavior (learning rate) based on the bucket and bag prediction error on the previous trial. Agents trained with a larger discount hyperparameter ($\gamma = 0.95$) show higher learning rates as prediction errors increase in the change-point condition (orange) and smaller learning rates as prediction errors increase during the oddball condition (brown). Conversely, the agent trained with a smaller discount hyperparameter ($\gamma = 0.5$) exhibits under-updating behavior during the change-point and over-updating behavior during the oddball condition (Nassar et al., 2021).

### RNN hyperparameters influence decision-making

To study how the hypothesized mechanisms influence the bag drop prediction and bucket update behavior, we trained agents with different hyperparameter combinations where each combination was initialized with 50 different seeds. Each agent's weights were fixed after training and their bucket update behavior was sampled over 10 epochs of 200 trials in both change-point and oddball conditions. Figure 2 depicts the area under the learning rate versus prediction error curves for the change-point (orange) and oddball (brown) condition curves as a function of the four hyperparameters. The difference in the area under the change-point curve and the area

under the oddball curve ($\Delta$ Area, black) indicates the magnitude of different behaviors during both conditions. Specifically, a smaller $\Delta$ Area suggests three possible scenarios where the agent could be under-updating during the change-point, over-updating during the oddball, or both.

Varying the reward discount hyperparameter ($\gamma$) led to a non-monotonic change in $\Delta$ Area, as previously seen in Kumar et al. (2022). Increasing the hyperparameter from 0.1 to 0.8 increases the learning rate for the change-point condition at a faster rate than for the oddball condition. Increasing discount factor to 0.99 caused a sudden decrease in change-point learning rate, due to credit assignment challenges, and unbounded value estimation causing numerical instabilities Sutton & Barto (2018). Near-optimal performance is observed for $\gamma \in [0.7, 0.95]$. Increasing the scale of the advantage learning signal ($\beta_\delta$) leads to a monotonic increase in $\Delta$ Area, where the change-point learning rate increases faster than the oddball learning rate. Alternatively, increasing the probability of resetting the recurrent dynamics with randomly sampled Gaussian noise causes a monotonic decrease in $\Delta$ Area towards zero with the change-point learning rate decreasing faster than the oddball learning rate. Lastly, increasing the replay buffer from 5 to 50 causes a monotonic increase in $\Delta$ Area with the change-point learning rate increasing faster than oddball. However, increasing the memory buffer size beyond 50 did not cause a further increase but a slightly decrease instead. This could be because a larger rollout length requires a larger number of trajectory samples before the model weights are updated, indicating sample inefficiency.

### Hyperparameters influence unstable fixed points

Next, we tested how hyperparameters influenced neural dynamics. Figure 3 shows example neural dynamics and fixed points found in two agents during the change-point and oddball conditions when trained with a large ($\gamma = 0.95$) and small ($\gamma = 0.5$) discount factor. The near-optimal agent ($\gamma = 0.95$) has a larger number of unstable fixed points (orange) for the change-point condition compared to the oddball condition. Conversely, the suboptimal agent ($\gamma = 0.5$) has a smaller number of unstable fixed points in both change-point and oddball conditions. This difference is robust across shuffle and eigenmode controls (Sup. Fig. 9).

Fig. 4 shows how the number of stable and unstable fixed points change for the change-point and oddball conditions when the four hyperparameters are varied over 50 seeds. The number of unstable fixed points increases non-monotonically when the reward discount hyperparameter ($\gamma$) and memory buffer ($\tau$) are increased, monotonically increase with a higher Advantage learning signal scaling ($\beta_\delta$) and monotonically decrease with a higher incidence of resetting the RNN dynamics ($p_{reset}$). Interestingly, the number of stable fixed points for both the change-point and oddball conditions vary at the same rate when changing each hyperparameter, causing the difference in stable fixed points to be consistent or close to zero (Sup Fig. 7).

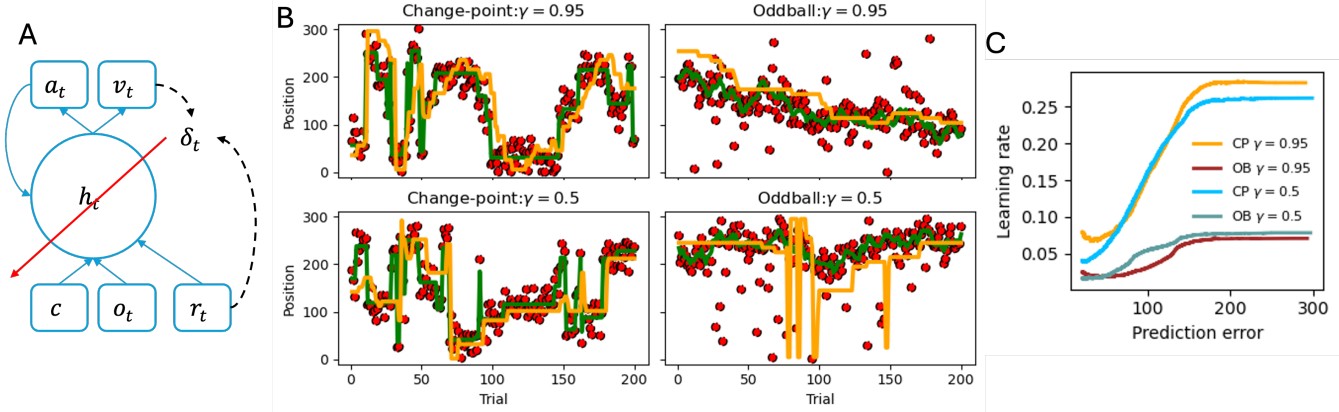

Figure 1: **Under and over-updating behaviors in agents trained with a smaller reward discount factor (γ). (A)** Agent architecture: The agent receives inputs including the task condition (change-point or oddball), observations (bucket location, bag drop location, and the previous trial's prediction error), and the reward at each timestep. These inputs are processed by a recurrent neural network (RNN), whose activity is passed to an actor and a critic. The actor learns the decision-making policy, while the critic learns the value function. All agent parameters are optimized using the Advantage function. **(B)** Two example agents (Top: $\gamma = 0.95$, Bottom: $\gamma = 0.5$) learn to move the bucket (yellow trace) to track the hidden helicopter's location (green trace) based on the evidence collected from bag drop locations during the change-point (left) and oddball (right) conditions. While both agents can track the helicopter's location, the agent trained with $\gamma = 0.5$ exhibits under-updating in the change-point condition and over-updating in the oddball condition. **(C)** The optimal strategies for the change-point and oddball conditions are to increase or reduce bucket displacement (learning rate) in response to large prediction errors, respectively. However, a smaller reward discount factor causes the agent to deviate from these strategies, leading to under-updating in the change-point condition and over-updating in the oddball condition. This behavior mirrors the paradoxical updating patterns observed in schizophrenia patients. Learning curves were generated by pre-training two agents with different reward discount factors and sampling 100 epochs of 200 trials each, with fixed agent parameters.

## Behavior is influenced by unstable fixed points

Figure 5 shows the correlation between the difference in change-point and oddball condition behavior (Δ Area) and the number of unstable fixed points found in the recurrent dynamics. A higher Δ Area value is indicative of a near-optimal bucket update behavior. The number of stable fixed points did not change with behavior suggesting that the number of stable fixed points is task dependent (Sup. Fig. 7), hence we do not see a similar positive correlation between behavior and the number of stable fixed point attractors Sup. Fig. 8.

The correlation between Δ Area and Δ unstable fixed points remains significantly positive for the reward discount ($\gamma$), scaling ($\beta_\delta$) and rollout ($\tau$) hyperparameters suggesting that the fixed point structure is moderately ($0.1 < R < 0.2$) but significantly predictive ($p < 0.05$) of task performance. One hypothesis that may explain this is that near-optimal agents use unstable directions in the vicinity of unstable fixed points to selectively amplify inputs in the changepoint condition, but not in the oddball condition. Hence, we analyzed the cosine similarity between each unstable fixed point's leading Jacobian eigenvector and observed state updates for nearby hidden states (Fig. 9). The heavy-tailed distribution shows that a number of unstable fixed points bias nearby trajectories along their unstable directions, and this effect is hyperparameter de-

pendent. Hence, some unstable fixed points actively shape task-relevant computations, suggesting their functional role in the agent's performance.

Fixed point identification adds Gaussian noise ($\xi_t \sim N(0, 0.5)$) to hidden states during search without propagating it over time, whereas behavioral evaluation injects persistent noise via $p_{reset}$ during task performance. Thus, each method perturbs the system differently—affecting local stability versus sequential processing—which explains the near-zero correlation for the $p_{reset}$ hyperparameter.

## SCZ patients demonstrate complex updates

Thus far, we have analyzed the behavior of artificial RL agents using the Δ Area metric, where a larger value indicates a near-optimal bucket update behavior for the change-point and oddball conditions. To better understand the relevance of this metric, we next compared the behavior of suboptimal RL agents to that of human participants. Figure 6A shows learning rate versus prediction error curves for healthy control subjects ($N_{control} = 32$) and patients diagnosed with schizophrenia or schizoaffective disorder ($N_{patients} = 102$) during the same predictive inference task (Nassar et al., 2021). Two key phases emerge in which update behavior differs between healthy controls and patients. First, when considering the area

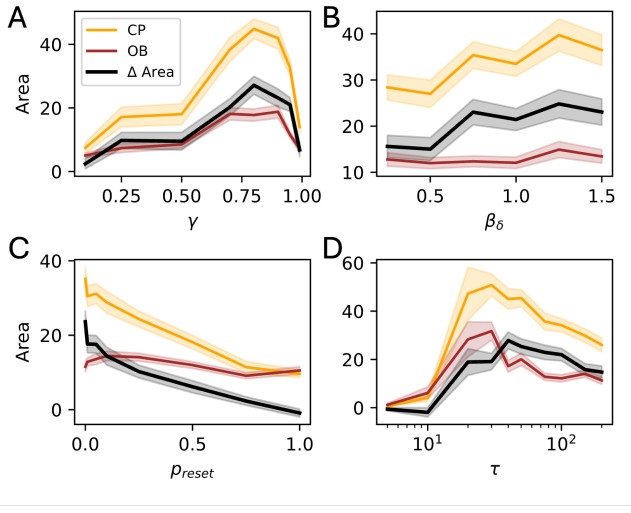

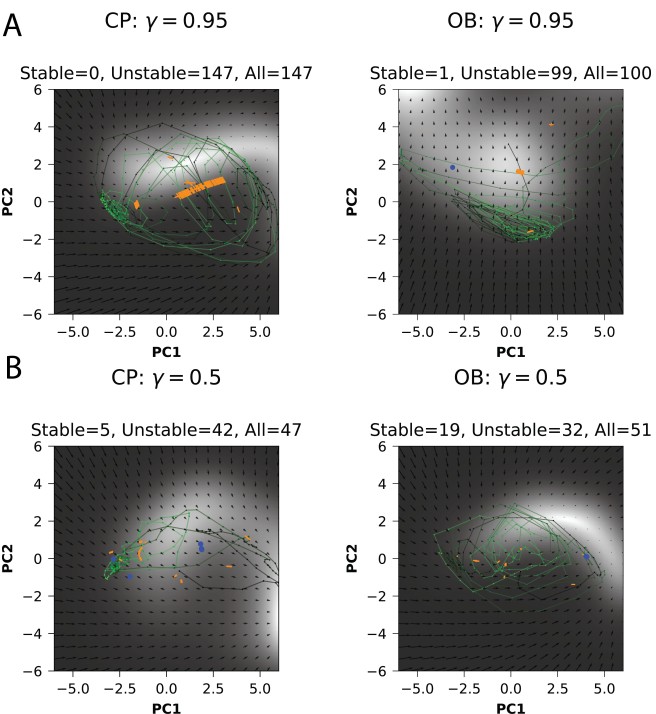

Figure 2: **Hypothesized mechanisms influence agents' decision-making behavior.** 50 agents with different seeds were trained by varying each parameter ($\gamma$, $\beta_\delta$, $p_{reser}$, $\tau$). For each agent, a learning rate versus prediction error curve (similar to Fig. 1C) was computed by sampling 10 epochs of 200 trials each. The area ($A$) under the learning rate curves for the change-point (CP, orange) and oddball (OB, brown) conditions was calculated for each parameter value. The black line shows the difference in area between the CP and OB conditions ($\Delta$ Area, Eq. 8). A higher value in the black line indicates closer to optimal decision-making behavior, where agents update their bucket position to match the magnitude of the prediction error in the CP condition but refrain from updating in the OB condition. **(A)** Increasing the reward discount factor ($\gamma$) results in non-monotonic decision-making behavior, with near-optimal performance observed $\gamma \in [0.7, 0.95]$. **(B)** Increasing the scale of the Advantage function leads to a monotonic improvement. **(C)** Increasing the probability of resetting the RNN dynamics leads to a decrease in decision-making performance, with a faster decrease in the area under the change-point learning rate curve. **(D)** Increasing the rollout size causes a non-monotonic decision-making behavior with near-optimal performance observed for $\tau \in [20, 100]$. Shaded area is 95% CI for 50 seeds.

under the learning rate curves for prediction errors greater than 20 (Fig. 6B), patients significantly under-update in the change-point condition ($t = -2.3, p = 0.02$) and show a non-significant trend toward under-updating in the oddball condition ($t = -1.4, p = 0.174$), with $\Delta$ Area not significantly different between the two groups ($t = -0.61, p = 0.545$). When restricting the analysis to prediction errors between 20 and 95 (Fig. 6C), patients significantly under-update their bucket position in both the change-point ($t = -2.2, p = 0.029$) and oddball ($t = -2.1, p = 0.036$) conditions compared to controls. For large prediction errors (greater than 95; Fig. 6D), patients under-update in the change-point condition ($t = -1.5, p =$

Figure 3: **Suboptimal agents exhibit fewer unstable fixed points.** Example neural dynamics in near-optimal (top, $\gamma = 0.95$) and suboptimal agents (bottom, $\gamma = 0.5$) in change-point (left) and oddball (right) conditions. Black to green lines show the evolution of state trajectories across time. Blue and orange markers indicate stable and unstable fixed points found by the fixed point finder algorithm. Flow fields represent velocity in the top two PCs, whereas lighter regions correspond to slower speed. The near-optimal agent has more unstable fixed points for both change-point and oddball conditions than the suboptimal agent.

$0.147$) and over-update in the oddball condition ($t = 1.1, p = 0.279$) relative to control subjects. The further breakdown of this pattern better mirrors the paradoxical under- and over-updating behavior previously described in schizophrenia. Notably, patients also exhibit a significantly lower $\Delta$ Area compared to healthy controls ($t = -2.0, p = 0.047$), a pattern that aligns with the performance differences observed between near-optimal and suboptimal RL agents. These findings highlight two key considerations. First, quantifying bucket update behavior specifically in the high prediction error regime may better capture the distinct over- and under-updating behaviors observed in SCZ. Second, the $\Delta$ Area metric offers an approach to characterize deviations from optimal updating, providing a framework for understanding decision-making impairments in psychiatric populations.

## Discussion

The present study integrates recurrent neural networks (RNNs) with reinforcement learning (RL) to investigate how

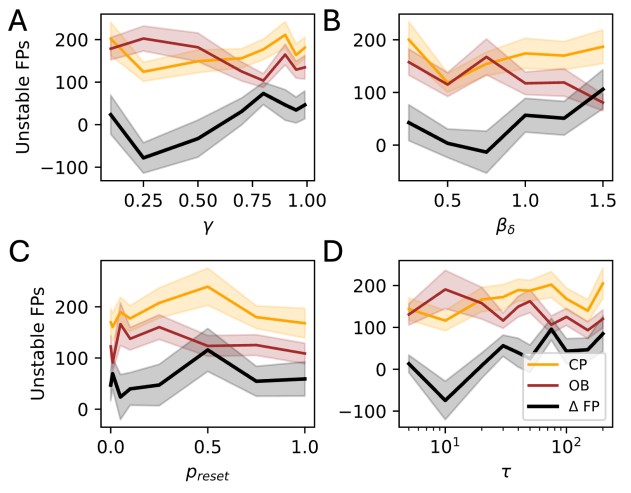

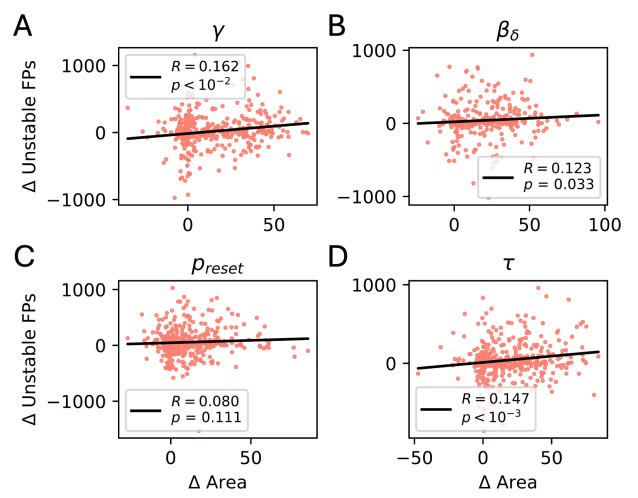

Figure 4: **Hypothesized mechanisms influence the number of unstable fixed points.** 50 agents initialized with different seeds were trained by varying each parameter ($\gamma$, $\beta_\delta$, $p_{reset}$, $\tau$). Within a 95% CI (shaded area), the number of unstable fixed points for the change-point (CP, orange) and oddball (OB, brown) conditions was found using the fixed point finder algorithm for each parameter value. The black line represents the difference in the number of unstable fixed points between the CP and OB conditions ($\Delta$ FP). **(A)** Increasing the reward discount factor ($\gamma$) results in non-monotonic change in unstable fixed points. **(B)** Increasing the scale of the Advantage function leads to a monotonic increase. **(C)** Increasing the probability of resetting the RNN dynamics leads to a decrease in unstable fixed points. **(D)** Increasing the rollout size causes a non-monotonic change in unstable fixed points. For stable fixed points, refer to Fig 7.

Figure 5: **Decision making behavior is correlated with the number of unstable fixed points.** The scatter plots show the correlation between $\Delta$ Area and $\Delta$ unstable fixed points, which describes the difference in area, $\Delta A$ and the number of unstable fixed points between the change-point and oddball condition. A larger $\Delta$ Area suggests a near-optimal performance. A positive correlation is observed when varying all the hypothesized mechanisms, suggesting a form of dependency between the number of unstable fixed points and decision-making behavior. The number of points in each plot includes 50 seeds multiplied by the variations in the parameters. The correlation between $\Delta$ area and $\Delta$ stable fixed points is close to zero. Refer to Fig. 8.

latent state inference and neural network dynamics contribute to decision-making deficits. By leveraging a predictive inference task, we examined how RNNs can learn to behave near-optimally, sub-optimally, and develop internal task representations over time. Specifically, we explored how four hyperparameters ($\gamma$, $\beta_\delta$, $p_{reset}$, $\tau$) influence behavior and the number of fixed points learned by agents. We also showed a positive correlation between the number of fixed points and update behavior, suggesting an interdependence between network dynamics and decision-making behavior. Lastly, a comparison between RNN-RL agents and human subjects elucidated how sub-optimal agent behavior might map onto behavior in patients and healthy controls.

Using SCZ as a case study, our results suggest that RL agents can be trained with RNNs to model behavioral deficits and can provide insight into their underlying neural dynamics. While frameworks such as the Hierarchical Gaussian Filter or model-based RL have been instrumental in modeling latent state inference (Mathys et al., 2011; Adams et al., 2018), they require strong a priori assumptions about task

structure and learning rates. They often assume a singular latent stateGershman et al. (2010) and may overlook cognitive strategies employed in decision-making, which can be uncovered using high-dimensional approaches (Ji-An et al., 2023). Our RNN-RL approach, in contrast, learns both state representations and policy mappings directly from experience, enabling discovery of emergent latent representations and associated neural dynamics without requiring a predefined model structure or strict parametric constraints (Doerig et al., 2023). Prior work suggests that SCZ-related cognitive impairments arise from deficits in inferring task structure (Nour et al., 2024; Hauke et al., 2024), with symptoms like delusions and hallucinations explained by disruptions in latent state formation (Erdmann & Mathys, 2022; Benrimoh et al., 2019). Our findings extend this view by demonstrating that impairments in structure formation—rather than reward sensitivity—can also account for sub-optimal performance.

From a dynamical systems perspective, psychiatric disorders have been conceptualized as disruptions in attractor dynamics (Scheffer et al., 2024; Roberts et al., 2017). Unstable fixed points have been proposed as a key feature of aberrant belief updating (Adams et al., 2018) and align with behavioral rigidity in SCZ (Adams et al., 2022). Our results

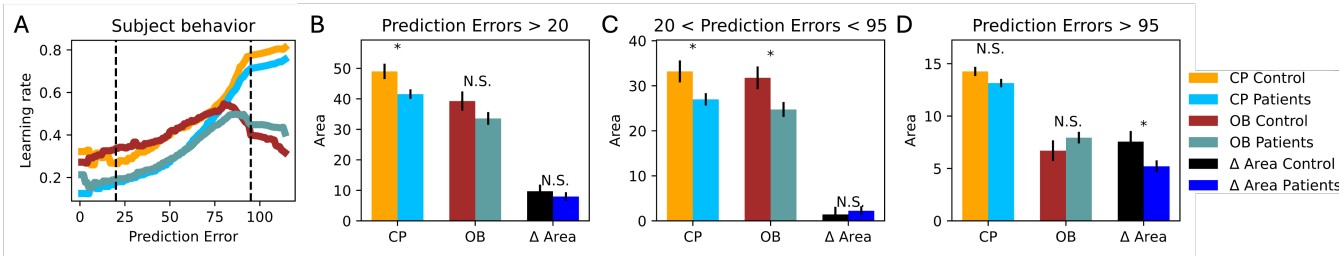

Figure 6: **Humans demonstrate complex update behavior in change-point and oddball conditions. (A)** The learning rate versus prediction error curves by control and patient data collected from Nassar et al. (2021) during the change-point and oddball conditions is visualized. Different update behaviors are observed with different prediction errors. **(B)** When we consider the area under the learning curves for prediction errors greater than 20 ($PE > 20$), patients show a significant under-updating behavior compared to controls during the change-point condition ($p < 0.05$). However, patients do not show a statistically significant over-updating behavior during the oddball condition ($p = 0.174$). The difference in area between the change-point and oddball conditions is also not significantly different ($p = 0.545$). **(C)** When we consider the area between prediction errors 20 and 95, patients tend to significantly under-update in both change-point and oddball conditions ($p < 0.05$). **(D)** Only when we consider the area greater than prediction error of 95, we start to see a resemblance of the under-updating behavior during the change-point and over-updating behavior in oddball conditions. However, these are not statistically significant ($p > 0.05$). Interestingly, patients show a statistically smaller difference in the area between the change-point and oddball curves ($\Delta$ Area, $p < 0.05$) compared to healthy patients, resembling the suboptimal behavior predicted by the normative model described in Nassar et al. (2021) and our RL agents. Hence, $\Delta$ Area can be an additional metric to analyze and classify suboptimal decision-making in the predictive inference task. Statistical tests were performed using independent 2-sided t-test.

link unstable fixed points to task performance, suggesting that deficits in latent state inference may be driven by network dynamics preventing efficient updating in response to changing environmental contingencies. Notably, prior work has shown that even in randomly connected RNNs, unstable fixed points shape network dynamics by acting as "partially attracting landmarks" (Stubenrauch et al., 2023). Future work should assess whether hidden state updates near these fixed points correspond to specific eigenmodes, allowing for rapid belief updating in response to select inputs.

One key area for future exploration is the evolution of RNN dynamics across training and learning phases, a shift that mimics real-world cognitive challenges. Analyzing changes in network representations between these phases will provide a deeper understanding of how stable and unstable fixed points emerge (Lin et al., 2023). Notably, the parameters $p_{reset}$ and $\tau$ control model-based components of the network, potentially revealing mechanistic insights into SCZ-related impairments in latent state formation. After manipulating parameters directly built into the critic ($\gamma$, $\beta_\delta$), fixed point differences arose in the RNN, which parameterizes the associated actor. In this way, the meta-RL architecture allows us to separate model-based and model-free components, akin to separations in predictive coding(Wang et al., 2018; Sterzer et al., 2018).

Several limitations of this approach should be noted. Particularly, the behavioral metrics to evaluate the performance of the RL agents could be improved. The area between the changepoint and oddball curves ($\Delta A$) offered one promising approach, but likely is not fully sufficient for characterizing behavioral phenotypes, particularly in human data, where asym-

metric sensitivity to prediction errors complicates performance across conditions. More nuanced metrics will be needed to classify optimal behavior in RNNs and compare them with human behavioral patterns (Nassar et al., 2021), especially in patient populations. Future work will formally fit Bayesian models to both patient and RNN-RL agent behavioral trajectories, which would provide insight into how RNN dynamics correspond to interpretable, Bayesian model parameters.

By grounding RL agents in empirical findings, we establish a benchmark for interpreting emergent latent neural representations. Furthermore, the present predictive inference task was chosen to target one specific behavioral cognitive deficit, necessitating validation across a battery of cognitive tasks. This RNN-RL framework is broadly applicable and should be adaptable to investigate cognitive dysfunction across other psychiatric populations. By varying task structure and reward contingencies, the same architecture could be retrained on (more naturalistic) paradigms targeting symptoms relevant to depression, obsessive-compulsive disorder, or anxiety disorders (Huys et al., 2016; Gradin et al., 2011). Moreover, task-specific hyperparameter tuning may help reveal transdiagnostic versus disorder-specific mechanisms in belief updating (Yang et al., 2019; Cloos et al., 2024; Rhoads et al., 2024).

By bridging computational neuroscience, computational psychiatry, and dynamical systems theory, this work underscores the potential of RNN-based RL models in elucidating mechanisms of cognitive dysfunction. Future research should continue to refine network architectures and inference frameworks to better capture the complex interplay between latent state formation, cognitive biases, and psychiatric symptoms.

## Acknowledgments

We thank the organizers and funding agencies of the Analytical Connectionism Summer School (2024), for which this work was begun by a group collaboration during the school. MGK and CP are supported by NSF Award DMS-2134157. AM is supported by the Monash University MITS and MGS scholarships. CP is further supported by NSF CAREER Award IIS-2239780, DARPA grant DIAL-FP-038, a Sloan Research Fellowship, and The William F. Milton Fund from Harvard University. SR is supported by the NIH Director's Early Independence Award DP5OD037383. This work has been made possible in part by a gift from the Chan Zuckerberg Initiative Foundation to establish the Kempner Institute for the Study of Natural and Artificial Intelligence.

## Code Availability

The code to generate all the figures can be found at `https://github.com/mgkumar138/subopt_rnn_rl`.

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

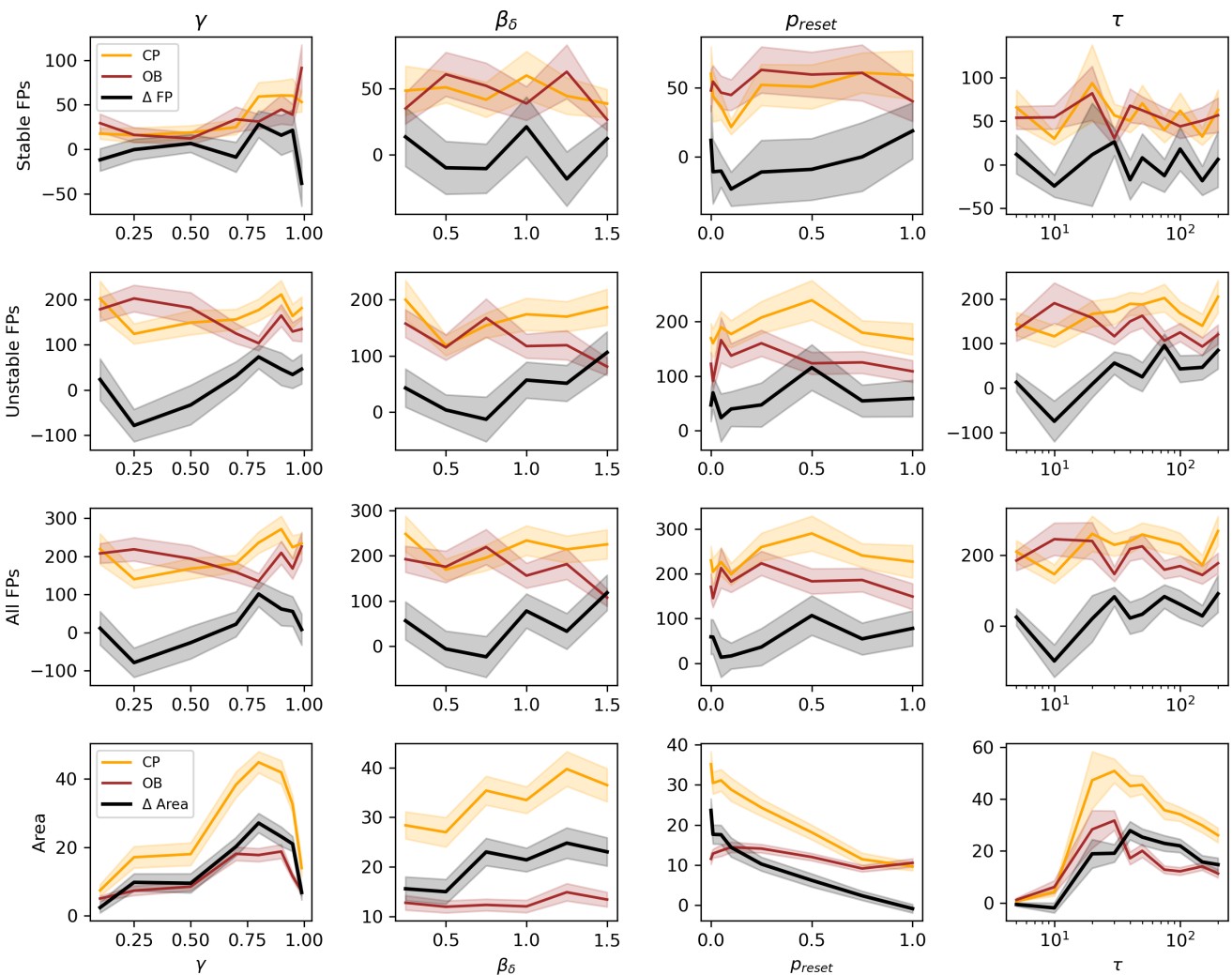

Figure 7: **Impact of Hyperparameters on Fixed Points and Agent Decision-Making Behavior.** Altering each hyperparameter results in a similar rate of variation in the number of stable fixed points for both change-point and oddball conditions, leading to a negligible difference in the number of stable fixed points between these conditions. In contrast, the number of unstable fixed points exhibits significant variation across different hyperparameter settings. This variation in unstable fixed points mirrors the changes in the area under the learning rate versus prediction error curves, indicating a comparable rate of change

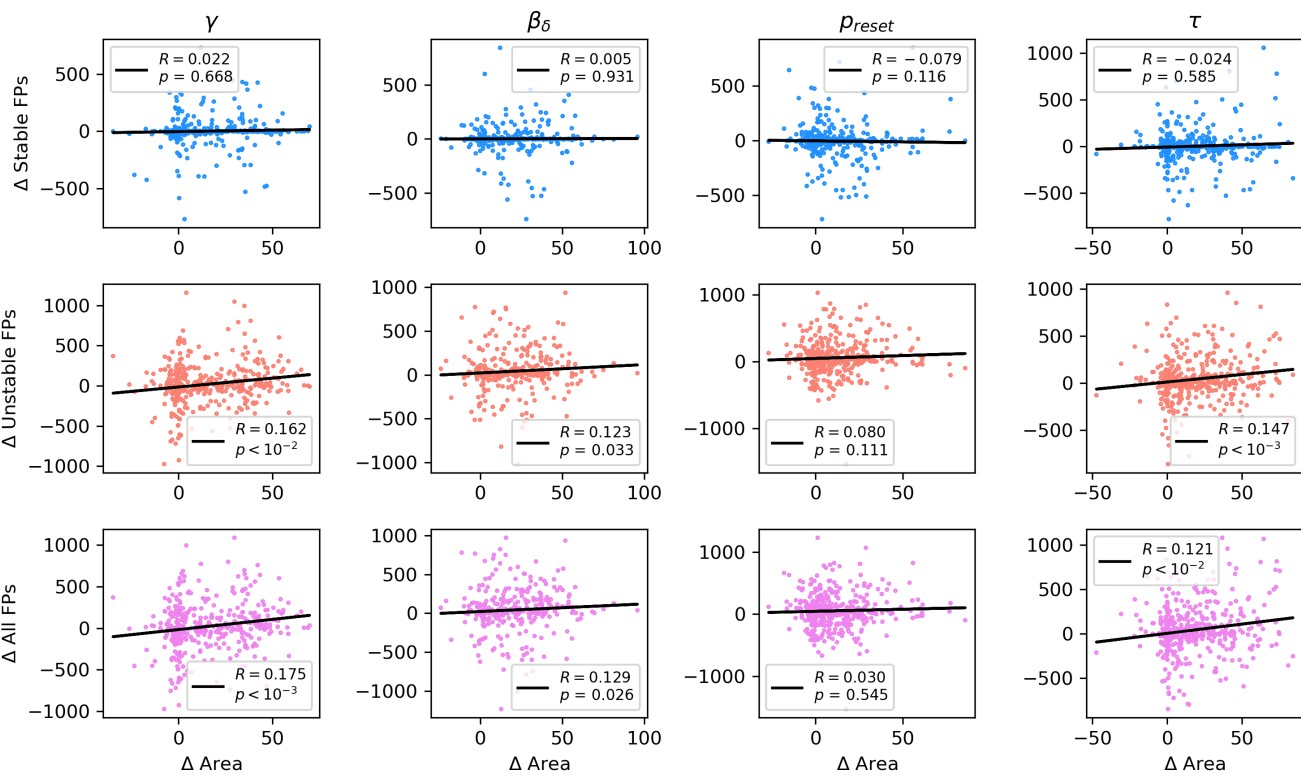

Figure 8: **Correlation Between Number of Fixed Points and Behavioral Optimality.** $\Delta$ Area represents the difference in the area beneath the curves for the change-point and oddball learning rates. A higher $\Delta$ Area signifies near-optimal behavior, characterized by proportional updates in response to prediction errors during change-point conditions and stable bucket positioning during oddball conditions. When hyperparameters are varied, the correlation between the number of stable fixed points and $\Delta$ Area approaches zero, indicating that stable fixed points do not significantly influence decision-making behavior. Conversely, the number of unstable fixed points shows a positive correlation with $\Delta$ Area, suggesting interdependence between network dynamics and behavior optimality.

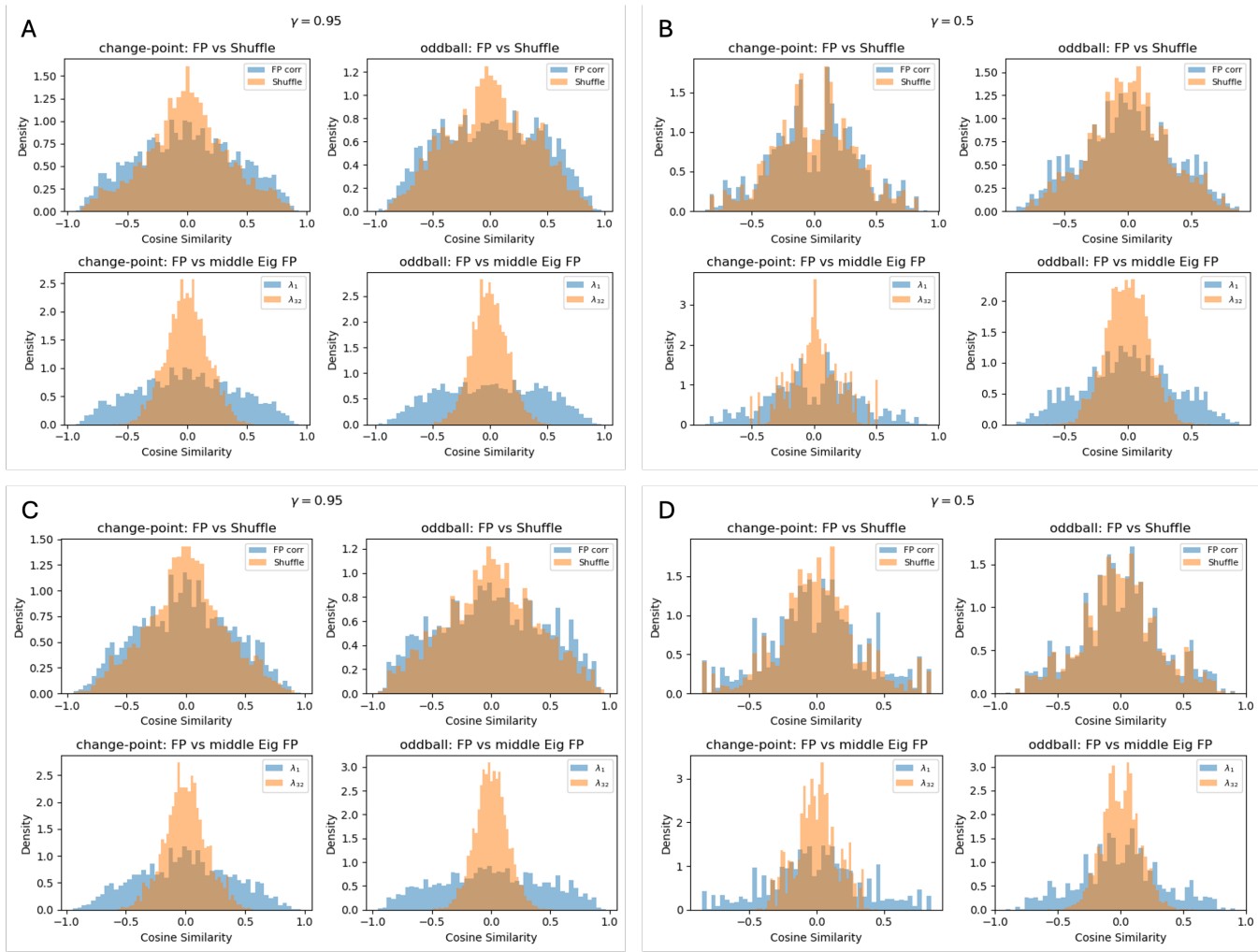

Figure 9: **Unstable modes associated with unstable fixed points shape observed RNN dynamics, depending on task context and reward discount factor (γ).** For each unstable fixed point ($u^*$), we identified hidden states within a tolerance radius ($\|u^* - h_t\| < 2$) and computed the cosine similarity between the leading Jacobian eigenvector ($v_1^*$) and subsequent state updates ($h_{t+1} - h_t$). **(A, B)** Cosine similarity distributions between the leading Jacobian eigenvector ($v_1^*$) of unstable fixed points ($u^*$) and actual state updates ($h_{t+1} - h_t$) within the vicinity of $u^*$ across all trajectories. Unstable modes in networks trained with high γ (0.95, **A**) show heavier-tailed similarity distributions (excess kurtosis ≈ −0.84 to −0.91) than controls (shuffled states: −0.21 to −0.37; median eigenmode: 0.0 to −0.2), indicating stronger dynamical influence. Low γ (0.5, **B**) yields lighter tails (kurtosis ≈ −0.59 to −0.85) closer to controls (shuffled states: 0.0 to −0.5; median eigenmode: −0.13 to −0.24), suggesting weaker fixed-point effects. **(C, D)** Analysis restricted to states only when a change-point or oddball hazard occurs. High γ (**C**) preserves heavy tails (kurtosis ≈ −0.70 to −0.89) than controls (shuffled states: −0.2 to −0.5; median eigenmode: 0.0 to 0.17), while low γ (**D**) further diminishes fixed-point influence (kurtosis ≈ −0.1 to −0.23) closer to controls (shuffled states: 0.0 to 0.7; median eigenmode: 0.0 to −0.32), also suggesting weaker fixed-point effects. Density includes fixed points found over 50 RNNs initialized with random seeds.

