# OpenReview forum: "Neurocomputational Underpinnings of Suboptimal Beliefs in Reinforcement Learning Agents"
_ccneuro.org/CCN/2025/Proceedings — CCN 2025 Proceedings asProceedingsPoster_

### Official Review · Reviewer_ND94 · 2025-03-26
**The research article explores the neurocomputational underpinnings of suboptimal belief updating in reinforcement learning (RL) agents, using schizophrenia (SCZ) as a test case. It employs a recurrent neural network-based RL framework to investigate decision-making impairments observed in SCZ, specifically under-updating in volatile environments and over-updating in response to uninformative cues. The study manipulates key hyperparameters—reward prediction error discounting, RPE scaling, working memory function, and episodic memory capacity—to model SCZ-like cognitive deficits. The results suggest that suboptimal agents exhibit fewer unstable fixed points, indicating reduced computational flexibility. This work advances computational psychiatry by linking cognitive biases to neural dysfunction and demonstrating how RL-driven RNN models can mimic psychiatric decision-making impairments.**

**Soundness:** 2
**Clarity:** 2

**Comments:**

Suggestions for Improvement
Clarity in Methodological Details

The study manipulates multiple hyperparameters to model SCZ-like behavior, but more clarity on the specific values chosen for these parameters and their real-world relevance would improve reproducibility.

Providing hyperparameter sensitivity analysis could help clarify how robust the findings are to small changes.

Comparison with Alternative Models

The paper primarily compares its approach to Bayesian and RL-based models but lacks a thorough discussion on how it performs relative to other existing computational psychiatry frameworks (e.g., Hierarchical Gaussian Filters, Bayesian inference models).

Including a comparative analysis would strengthen the claim that the RNN-RL model provides novel insights.

Empirical Validation with Human Data

While the study compares agent behavior to patient data from Nassar et al. (2021), a more quantitative comparison, such as model fitting to patient behavior, would solidify the findings.

The study could benefit from cross-validation using additional datasets from other SCZ studies to verify generalizability.

Broader Applicability to Other Psychiatric Conditions

The focus is on schizophrenia, but could the model generalize to other psychiatric disorders with decision-making deficits, such as depression or OCD?

Including a discussion on how the framework could be adapted for other conditions would broaden its impact.

Interpretability of Neural Network Dynamics

The analysis of fixed points is intriguing, but the biological interpretation of these computational findings could be expanded.

Connecting these network dynamics more explicitly to neural circuits or brain regions affected in SCZ (e.g., dopaminergic pathways, prefrontal cortex dysfunction) would enhance the neuroscientific relevance of the results.

By addressing these areas, the paper can further strengthen its contributions to computational psychiatry and make its findings more applicable to both neuroscience and clinical practice.

**Expertise:**

2

**Interest:**

3

---

> ### Author Rebuttal · Authors · 2025-04-15
>
> >Hyperparameter analysis
>
> We conducted systematic analyses of how each hyperparameter (learning rates, memory capacity, etc.) impacts both behavior (Fig. 2) and Fixed Points (Fig. 4). The 50 random seeds demonstrate robustness to initialization.
>
> >Model comparison
>
> We modified the discussion for comparison (Please see lines 488/493)
>
> >Normative model fitting
>
> We agree that future work should incorporate normative model fitting to the RNN-RL agent behavior (please see response to Reviewer 2), to bridge our model with existing frameworks and clarify their shared and divergent assumptions (Please see line 547).
>
> >Generalize to other disorders
>
> We have added to the Discussion on the model's generalizability and flexibility (Please see line 556).
>
> >Interpretation of computational findings
>
> Using fixed-point analysis (Sussillo & Barak, 2013), we found optimal agents maintain more unstable fixed points than suboptimal agents (despite equal totals), mirroring cortical decision circuits where unstable points enable flexibility (Mante et al., 2013) and neural systems' stability-flexibility tradeoffs. This suggests SCZ may involve disrupted state transitions through similar dynamical impairments. We refer to comments to reviewer 2 on the extent of biological plausibility inferred here,  and note that this approach offers concrete ways to study cognitive dysfunction through dynamical systems theory.
>
> >Network dynamics to neural circuits
>
> We clarify our model's neural circuit mappings to mirror three SCZ-relevant systems:
> PFC-like computation: The RNN maintains working memory and belief states, capturing PFC's integrative role (Wang et al. 2018; Yang et al. 2019) and its SCZ-related dysfunction (Sigurdsson & Duvarci, 2016). Altered RNN dynamics under different hyperparameters reproduce characteristic SCZ working memory deficits (Braun et al., 2021).
> Striatal action selection: The actor-critic module reflects basal ganglia function, with RPE updates modeling dopaminergic signaling (Schultz, 2016). This is particularly relevant given SCZ's dysregulated cortico-striatal loops (Howes et al., 2017).
>
> While abstracted, these mappings connect established SCZ pathophysiology (dopamine dysfunction, PFC-striatal miscommunication), explain how hyperparameter changes produce SCZ-like behaviors and generate testable hypotheses about circuit dysfunction (e.g., unstable fixed points ≈ PFC state transition failures). We expand discussions of these biological parallels in revision.

---

### Official Review · Reviewer_7ghF · 2025-03-30
**A gap between the neural dynamics of RNNs and human brains**

**Soundness:** 1
**Clarity:** 2

**Comments:**

This study proposes a RNN model using RL to evaluate the behavioral deficits of adaptations in volatile environments in schizophrenia. By altering the hyperparameters of the RNN models, they showed some similar adaptation deficits as in schizophrenia. However, as far as I understand, in my view, this approach cann’t provide mechanical insights on deficits in schizophrenia. The key gap is that the neural dynamics of RNNs and human brains in normal humans and patients are incomparable. For instance, in biological brains, mutual inhibitions between neurons play significant contributions on the dynamics of the brain system, as well as in schizophrenia. However, the RNN models in the current study lack such a key feature. The RNN models can redundantly simulate any input-output relationships by minimizing the loss functions. Behavioral similarities do not necessarily reflect the similarity of neural dynamics. I am wondering why the authors did not separately simulate the normal humans and patients’ behaviors and then compare their different neural dynamics with different parameters; Or finetuneing the RNN model from the normal subjects to patients. In belief, I am not convinced that the current study provides meaningful insights on neural mechanisms of schizophrenia.

**Expertise:**

3

**Interest:**

1

---

> ### Author Rebuttal · Authors · 2025-04-15
>
> >Mechanistic insights
>
> The reviewer raises a valid concern regarding the biological plausibility of the RNN-actor-critic model and its ability to provide mechanistic insights into schizophrenia (SCZ). We agree that RNN neural dynamics and biological brains are not directly comparable (e.g. absence of mechanisms like mutual inhibition). Our goal was to develop a minimal model to replicate behavioral patterns observed in SCZ. To do this, we had to abstractly represent reward prediction error computation, working memory, and episodic memory, systematically assess how these hyperparameters influence behavior using reinforcement learning as a normative framework (Series, 2020; Montague et al., 2012).
>
> Though these hyperparameters are not direct biological (nor cognitive) mechanisms, they allow us to explore how suboptimal behavior arises when key computational processes are perturbed. Our findings show that behavior previously reported in SCZ patients can be reproduced within this model via a shorter reward prediction error timescale and reduced episodic memory capacity. This implicates necessary abstract mechanisms underlying a broader hypothesis that impaired dopamine-driven reward estimation disrupts future decision-making—without claiming to identify specific neural mechanisms.
>
> Now, we can extend this model to a biologically plausible version to make predictions about neural mechanisms.
>
> >Separately simulating…fine tuning RNN model
>
> While behavior cloning (Aldarondo et al., 2024) would be valuable for comparing healthy and patient-specific dynamics, the available dataset (33 healthy/108 patients × 100 trials; Nassar et al., 2021) lacks both the granularity (time specific state updates versus trial-by-trial transitions) and scale needed to train RNNs without overfitting. More critically, supervised learning would: (1) limit our ability to study trial-and-error learning computations like explore-exploit tradeoffs and reward prediction errors that characterize the actual task, and (2) constrain emergent representations to human-specific behavioral patterns rather than the task's true latent dynamics. We therefore adopted the normative reinforcement learning framework (Singh et al., 2023; Kumar et al. 2024) which better captures the neuromodulatory computations underlying predictive inference while requiring fewer behavioral constraints. Future work with richer trial-level data should implement a cloning approach to directly compare model and human dynamics.

---

> > ### Comment · Reviewer_7ghF · 2025-04-22
> >
> > the authors adequately responded my comments. As the authors addressed, a great gap between the RNN model and SCZ remains to resolve.

---

### Official Review · Reviewer_mzZp · 2025-03-30
**Neurocomputational Underpinnings of Suboptimal Beliefs in Reinforcement Learning Agents**

**Soundness:** 1
**Clarity:** 2

**Comments:**

This work studies the neurocomputational basis of how maladaptive beliefs lead to behavioural abnormalities seen in patients diagnosed with schizophrenia related psychiatric disorders. The study focuses on reinforcement learning agents implemented with recurrent neural networks. The agent is trained on a specific task that has been previously used to access patients with schizophrenia, notably that these patients appear not to take into account sufficiently for evidence in volatile environments and react too much to irrelevant cues. Authors focus on several metaparameters in the model and track how these influence model performance and also how these affect the fixed points on the internal NN dynamics. Best I can tell the most important finding is that the hyperdiscounting significantly influences the behaviour in a way that is similar to what is seen in SCZ patients (extrapolating from the specific example task: inability to correctly integrate evidence needed to infer trend information and taking into account too strongly randomly dispersed evidence).

Authors then apply dynamical systems analysis to the trained RNNs and find that discounting impacts the number of unstable fixed points in the trained NN dynamics – hyperdiscounting decreases the number of fixed point thereby (hypothetically) impoverishing the network dynamics. Authors hypothesize that this is the mechanisms behind the patterns behavioural suboptimality. The study also includes a comparison of the model's behavioural metrics with human data from individuals with schizophrenia and healthy controls.

This is an interesting study that tries to do beyond a simple RL model of control and SCZ patient behaviour in that it links the patient suboptimality to the structure of internal representations in the neural networks that learns and encodes the behaviour. However, there are a number of points that should be addressed.


Points to address to improve the paper:

How was the number of the initial point for the fixed point evaluation picked? What is the rationale for 10 initial conditions? How exactly were these sampled? Can the authors state how and why this fixed point finder algorithm would find both stable (that is obvious) and the unstable fixed points? Or does it find stable fixed points and saddles by finding their stable manifolds?

What exactly does area under the learning rate vs the prediction error tell us? Needs to be better explained to a more general audience who are not RL experts

In general a more technical and detailed methods section would make the paper much clearer. As is we are sort of expected to guess how they did things specifically.

Authors note that there is a decrease of learning performance for agents that do not discount and speculate that it is due to  “This could be due to model collapse due to sparse rewards or numerical instability.” I think it is important for the authors to unpack this and to delve deeper into what is actually happening here. Surely some more analysis can be done?

A better link needs to be established between the number of unstable fixed points and performance. Meaning that in a model one can surely go beyond just showing a correlation between a smaller number of fixed points and decreased learning performance? The suggestion that unstable directions near these points might selectively amplify inputs in certain conditions requires further analysis. Can the authors study this?

A general methodological question – the authors focus on over discounting as a potential explanation of the learning pathology seen in SCZ patients. They draw a parallel between the RL agent results and the performance of the patients. This while promising does feel anecdotal. Why not try optimise the RL agents for the human control and SZ patient behaviour and “discover” what are the hyperparameters of such agents? Would that not give a much stronger result?

Authors used one metric for evaluating agent behaviour and comparing it to human data the  ∆ Area (see comment about asking for a better explanation what it measures exactly). While this metric quantifies differences in learning rates between conditions, the authors note its potential limitations in fully capturing the range of observed behavioural patterns, especially in human subjects exhibiting nuanced prediction error-dependent updates. This reliance on a single metric really limits the potential impact of this work as it restricts a comprehensive characterization of suboptimal behaviours.

**Expertise:**

2

**Interest:**

2

---

> ### Author Rebuttal · Authors · 2025-04-15
>
> >Sampling initial points for FP analysis
>
> Refer to line 290 in the revised PDF. We sampled 10 epochs of 200 trials (2000 initial states) each beginning with unique random seeds, generating independent hidden state trajectories. This sufficiently covered the state space (additional epochs didn't significantly change results).
>
> >Why the algorithm would find unstable FP
>
> The FP finding algorithm (Barak & Sussillo 2013) does not rely on iterating RNN dynamics and observing the limiting behavior. Rather, it directly optimizes a hidden state starting from an initial guess to minimize the norm of the one-step update ||h_{t+1} - h_{t}||^2 . It is necessarily true that all fixed points, stable and unstable, are at least local minima of this loss function, and thus can in principle be found.
>
> >Area under the curve
>
> Please refer to line 267 in the revised manuscript.
>
> >More detailed methods
>
> We will expand the methodology in the appendix and provide code to generate the results.
>
> >Decrease performance without discount
>
> With γ≈1 the value function becomes unbounded causing high variance, and difficult to assign credit to actions responsible for the reward (Sutton 1984; Arjona-Medina et al. 2019), especially in environments with sparse or noisy rewards (e.g. oddballs) (Sutton & Barto, 2018; Kearns & Singh 2000). Therefore, the optimal γ depends on the environment and needs to be tuned experimentally. Refer to line 351 in the revised manuscript.
>
> >Unstable FPs vs performance
>
> Our initial FP analysis lacked quantitative evidence linking these dynamical features to network function. We analyzed the cosine similarity between each unstable fixed points leading Jacobian eigenvector and observed state updates for nearby hidden states. The heavy-tailed similarity distribution shows that: (1) unstable fixed points systematically bias nearby trajectories along their unstable directions, (2) this effect is hyperparameter dependent, and (3) these dynamical features actively shape task-relevant computations rather than being epiphenomenal - confirming their functional role in the RNN's performance (See Fig. 9 in the revised PDF).
>
> >Optimize agents for human behavior
>
> Please see our response to Reviewer 2 Q 2.
>
> >Reliance on a single metric
>
> Future work will develop more nuanced metrics to study the different facets of suboptimal behavior e.g non & moderate updates in Nasser et al. 2021. Nevertheless, the single metric has been useful to discriminate subject behavior (Fig. 6).

---

> > ### Comment · Reviewer_mzZp · 2025-04-20
> >
> > the authors adequately addressed my comments.

---

### Meta-Review · Area_Chair_nMjB · 2025-05-04

**Ccn Recommendation:** Accept as Proceedings

**Metareview:**

The reviewers raised several important methodological concerns about this submission, including the need for more detailed fixed point analysis, questions about biological plausibility of RNN models for understanding neuropsychiatric mechanisms, and requests for more comprehensive model evaluation. I agree that those concerns were relevant for a paper making mechanistic claims about neural computation in schizophrenia. In their rebuttal, the authors addressed these concerns by explaining their fixed-point methodology, clarifying the scope of their biological interpretation, and acknowledging limitations of their evaluation metrics.

Overall, this submission makes a significant contribution by establishing a novel computational framework linking cognitive biases to neural dynamics in schizophrenia, with potential application to other psychiatric disorders. The revisions strengthened both the justification for the RNN-RL approach and the interpretability of the reported findings. Given the innovative approach connecting neural computation to observable behavioral deficits, the thorough response to reviewer concerns, and the potential impact on computational psychiatry, I recommend acceptance to the Proceedings.

**Summary:**

This submission received three reviews with varying evaluations. All three reviewers found the paper's approach of modeling decision-making in schizophrenia with RNN agents to be interesting, though opinions differed with respect to the soundness of the approach. The reviewers raised several methodological concerns including: (i) the need for more detailed fixed-point analysis methods; (ii) questions about the biological plausibility of RNN models for understanding neuropsychiatric mechanisms; and (iii) requests for more rigorous model evaluation and comparison with existing approaches. In their rebuttal, the authors made substantial revisions to address these concerns, including an expanded explanation of the methodologies, new analyses showing the functional role of unstable fixed points, clearer explanation of the limitations of the approach, and additional comparisons to other approaches in computational psychiatry. Two reviewers confirmed that their concerns were adequately addressed in the author's response.

**Expertise:**

3